# The Lived Experience of Gender and Gender Equity Policies at a Regional Australian University

**Jennifer Manyweathers [1,*]**, **Jessie Lymn [2]**, **Geraldine Rurenga [3]**, **Katie Murrell-Orgill [4]**, **Shara Cameron [5]** and **Cate Thomas [6]**

1 Graham Centre for Agricultural Innovation, Charles Sturt University, Locked Bag 588, Wagga Wagga, NSW 2678, Australia

2 School of Information Studies, Charles Sturt University, Locked Bag 588, Wagga Wagga, NSW 2678, Australia; jlymn@csu.edu.au

3 Division of Learning and Teaching, Charles Sturt University, Locked Bag 588, Wagga Wagga, NSW 2678, Australia; grurenga@csu.edu.au

4 School of Humanities and Social Sciences, Charles Sturt University, Locked Bag 588, Wagga Wagga, NSW 2678, Australia; kmurrell-orgill@csu.edu.au

5 CSU Engineering, Charles Sturt University, Locked Bag 588, Wagga Wagga, NSW 2678, Australia; shacameron@csu.edu.au

6 Faculty of Science, Charles Sturt University, Locked Bag 588, Wagga Wagga, NSW 2678, Australia; cthomas@csu.edu.au

**\*** Correspondence: jmanyweathers@csu.edu.au

**Abstract:** The research question driving this project was 'what is the congruence between the lived experience of gender and a policy designed to improve gender equity in a university environment?' The study used mixed methods to investigate the question. These methods included analysis of organisational travel data, and a collaborative autoethnography of participants engaging with claims for dependent care support expenses while travelling for work. The research found four key themes influencing the relationship between gender equity policies and the lived experience of staff. These include gatekeeping, organisation-wide funding of gender equity policies, policy development processes and gender equity as a concept. This article presents a series of transferable recommendations for organisations looking to improve gender equity.

**Keywords:** gender equity; work-related travel; dependent care; higher education

## 1. Introduction

The challenges that women face in the higher education sector relating to career advancement have been well documented (Morley 2013; Thomas et al. 2019). While much has been done to improve opportunities and raise awareness around the barriers that exist for women in both professional and academic roles in higher education institutions (David 2015), gender equity still has not been achieved. Research demonstrates that women continue to struggle to get access to and maintain careers in Science, Technology, Engineering and Mathematics (STEM) fields due to structural barriers (Science in Australia Gender Equity 2019). This is reflected in increases in the gender pay gap within the education and training sector (Workplace Gender Equality Agency 2020). and disadvantages society as a whole (Broadley 2015).

The Australian Government defines gender equality in the workplace through four lenses; equal pay for equal work; barrier-free access to participation in the workforce; access to all industries and roles, regardless of gender; and the 'elimination of discrimination on the basis of gender, particularly

in relation to family and caring responsibilities' (Workplace Gender Equality Agency 2020). This last lens forms the focus of this research investigation.

Gender equity strategies are key attempts by workplaces to address identified gender inequality. One key societal and organisational gender equity issue is the invisibility of the work of caring and the impact of this within educational institutions (Acker 2012; Grummell et al. 2009). Women continue to bear the burden of childcaring responsibilities. In Australia in 2016–17, women spent twice as long undertaking childcare duties each day than their male counterparts (Australian Government 2018). While this is a decrease from 2009, when women spent two-and-a half-times more time than men (Australian Government 2009), it is still a significant gender imbalance. The "Gendered Substructure" (Acker 2012, p. 215) that exists in organisations provides some explanation as to how gender inequality is maintained and reproduced through organisational processes. Without deliberate and strategic intervention from higher education institutions and the research sector, this systemic issue will continue (Grummell et al. 2009).

Reflection by academic organisations on their own cultural, social and economic processes that nurture inequality is required to avoid precluding anyone from fully engaging in their chosen careers based on their gender. This should include consideration of how the very practices of tenure and academic career advancement produce conformity by separating the jobs from the people who do them (Butterwick and Dawson 2005, p. 52; Eveline 2005, p. 643). By supporting expectations and performativity of university staff that can only be fulfilled by those without caring responsibilities, compliance to the non-reflexive, capitalist-driven institution will be maintained (Butterwick and Dawson 2005, p. 52; Grummell et al. 2009, p. 192). Without ongoing reflection, the gendered substructure will continue to restrict the valuable contribution by their staff to higher education, research and their community (Butterwick and Dawson 2005).

In the late 1990s, a regional Australian university developed a policy available to all staff, both professional and academic, to support those with caring responsibilities while they travelled for work. This was the only university at the time to recognise the need for such a policy and to support its staff in such a way. Currently, it is still one of only a few Australian universities that have implemented such a policy.

As part of academic institutional reflection, this study is based on the experiences and individual and collective reflections of university staff engaging with this unique policy within a single university. The policy was designed to assist caregivers, both men and women, with career advancement, by overcoming barriers associated with travel expectations. A collaborative autoethnographic approach was used in the study to reflect the lived experience of the research team as a group of women employed as both professional and academic staff by a regional university with distributed campuses, where travel for work is often required. The research question driving the project was, "what is the congruence between the lived experience of gender and a policy designed to improve gender equity in a university environment?"

The current policy aims to assist any member of staff, regardless of gender, with caring responsibilities to cover additional care-related expenses for any child under the age of 16, if the employee is required to travel for work-related reasons. This travel can include attending or presenting at conferences, or travel between university campuses. Successful applicants are entitled to receive up to $40 per day of travel towards dependent care expenses. The policy forms part of a broader university Travel Policy, and states that:

> ### Dependent Care Support
>
> *Staff who undertake work related travel are entitled to claim dependent care expenses up to forty dollars ($40.00) per day (not per child) for care that is over and above their normal dependent care cost arrangements. This applies for dependents up to the age of 16 years. Proof of age of the dependent may be requested.* (Anonymous 2019)

## 2. Materials and Methods

This study adopted a mixed methods approach, drawing upon both quantitative institutional data and a collaborative autoethnography to critically explore the lived experience of gender in a regional university setting, through the lens of a gendered policy. Using an autoethnographic design allowed the ethnographers to examine first the description of experiences relating to the policy and its implementation and then to analyse the power relations that exist to pervade these experiences, with the goal of real institutional reflexivity (Butterwick and Dawson 2005). The methods used included descriptive analysis of quantitative institutional data and journaling, conversations and observations, which are all discussed in more detail below.

All research activities involving data collection and analysis were approved by the host university's Human Research Ethics Committee (H19310).

### 2.1. Sample

There were a total of seven participants in the study; the five research team members and two other participants recruited through convenience sampling based on collegial relationships within the institution. Participant profiles are summarised in Table 1, noting that participants have diverse profiles. All participants had job descriptions that included an expectation that they would be available for travel. Because the study was considering the lived experiences, participants were identified by their institutional category to examine any similarities or differences across groups of academics versus professional staff, and full-time versus contract employees. All employees with caring responsibilities shared these responsibilities with partners. Four out of the five partners also worked full-time, while the fifth worked part-time and studied. Four of these five employees reported that the primary burden of childcare fell to them, rather than their partner.

**Table 1.** Participants in an autoethnographic study of lived experience of gender at a regional Australian university 2019.

| Participants' Profiles | Number of Participants with This Variable |
| --- | --- |
| Dependent care responsibilities for children | 5 |
| No dependent care responsibilities for children | 2 |
| Staff and budget management responsibilities | 2 |
| Long-serving staff members (10 years plus) | 3 |
| Recent (in the last 2 years) appointments to the university | 1 |
| Full-time contract | 4 |
| Contract employment | 3 |
| Academic staff | 5 |
| Professional staff | 2 |

### 2.2. Data Collection

#### 2.2.1. Institutional Data

De-identified institutional data about the Dependent Care Support expense claims ($40 per day payment) was obtained from the university's travel coordination office, along with the number of overall travel claims during the sample period, to establish a baseline for travel.

#### 2.2.2. Collaborative Autoethnography

Collaborative autoethnography provides a unique approach that enables the participants to be impacted by understanding and analysis of real life, individual experiences (Blalock and Akehi 2017). The methodological tools of writing, conversation and group analysis were used to share experiences and collaborate on building an understanding of the lived experience of the policy. Similar to autoethnography, collaborative autoethnography 'still focuses on self-interrogation but

does so collectively and cooperatively' (Chang et al. 2013, p. 21). This methodology was selected for its appropriateness to the framework the project was investigating—the lived experience of gender in a university setting. Lapadat argues that 'autoethnography offers fertile ground for interrogating dominant theoretical stances and hegemonic paradigms, and furthering social justice aims' (Lapadat 2017, p. 589), and is an appropriate method to investigate this field.

The research team used online and hard copy journaling to record their interactions, observations and reflections on their engagement with the policy. When talking about ethnography and the writing of culture and lived experience, Clifford and Marcus argue that:

> *To recognize the poetic dimension of ethnography does not require one to give up facts and accurate accounting for the supposed free play of poetry. Poetry is not limited to romantic or modernist subjectivism; it can be historical, precise, objective … Ethnography is a hybrid textual activity; it transverses genres and disciplines.* (Clifford and Marcus 1986, p. 26)

The team met weekly online and twice face-to-face to discuss and reflect on their journal entries and experiences. This included accidental and intentional interactions with other university staff members about the travel policy, captured in de-identified phone conversations and email exchanges. Notes from these collaborative analysis sessions were taken and, along with the journal entries, formed the data collected and analysed in this study.

## 3. Results

The de-identified institutional data highlights a policy that while not gender specific, is entirely used by women, albeit at a very low rate. Of the 18,959 approved staff travel claims from January 2018 until August 2019, just 13 (0.068%) included an approved Dependent Care Support expense payment. Table 2 shows the de-identified information about successful applications. Quantifying the success rate of dependent care allowance *applications* is not currently possible, as this data is not recorded. The lack of data about denied applications suggests that the institution is yet to establish a comprehensive procedure for applying for, processing and notifying applications about the allowance.

**Table 2.** Successful applications for Dependent Care Support, January 2018–August 2019.

| Date | Gender | $ Amount Paid |
|------|--------|---------------|
| 06/04 | F | 120 |
| 10/05 | F | 80 |
| 11/12 | F | 200 |
| 23/03 | F | 40 |
| 23/07 | F | 200 |
| 12/12 | F | 200 |
| 07/03 | F | 45.36 [1] |
| 29/06 | F | 80 |
| 29/08 | F | 120 |
| 08/08 | F | 120 |
| 25/02 | F | 120 |
| 12/02 | F | 200 |
| 20/05 | F | 40 |
| 01/08 | F | 80 |

[1] Unable to trace the significance of this amount.

Of note is that all expense payments over the survey period were made to women, and the payments ranged from $40 through to $200.

Narratives from the research team and other participants provided reflections on how the Dependent Care Support policy impacted their capacity to travel for work, both positively and negatively. Four main themes were developed from the data, including potential administrative and organisational barriers to policy uptake and how this policy might enhance gender equity

understanding and actions within the university. Any explanatory text within the data is included in square brackets. Any data removed for de-identification is noted (as such).

*3.1. Gatekeeping*

Participants reported experiencing variations of control, decision-making and gendered language relating to the application of the policy and included the following examples.

One participant discussed her claim for reimbursement being rejected by one level of administration, after travel was completed, even though the initial dependent care expense had been approved by the budget holder:

*"What I would like to point out—is that the form* [a form requiring applicant information has been periodically required] *includes provisions/conditions that are not part of the actual policy. I'm going to assume the policy was approved by a governance level process but that the form was created at the (University) local travel level.* (P7)

The same participant highlights the gatekeeping nature of policy implementation at universities and suggests that how the policy is implemented may raise other barriers that impact on the innovative benefits of such institutional support, including variable interpretation of entitlement.

*This is not the first time in a university I've been exposed to "extra conditions' made up by people at lower levels to governance on how a policy should be implemented. In this case of the travel policy, the form appears in direct conflict with the actual policy that only states proof of age is required—not any proof of additional costs.* (P7)

As participants engaged with the policy and included claims for dependent care expenses when travelling, multiple iterations of what documentation was required to claim expenses emerged through the data, revealing inconsistencies in processes and decision making. The following conversation occurred by email over an eight month period:

*We no longer require the form (Dependent Care Claim Form—see Appendix 3) to be completed. Staff need to include the request for the allowance in their on-line booking form. This is then included in the travel plan which is then approved by the Delegated Approver for that area. You may like to add the allowance is not paid in advance. Staff need to contact (University) Travel after the travel and request for the allowance to be paid.* (March 2019, email from (University) Travel)

*The form was reinstated on the 27th of September this year as there were some issues with staff questioning their ability to claim Dependent Care Allowance and additionally approvers willing to approve the expense.* (7th November 2019, email from (University) Travel)

*We have made a decision to not require the form to be completed effective immediately.* (14th November 2019, email from (University) Travel)

There were examples of staff claiming dependent care expenses and their success or otherwise depending on what sort of information they provided in their claim, and the language they used. The following interaction occurred early in 2019, and highlights the role of the administrative gatekeepers in determining what is considered 'normal' dependent care:

*I remembered the Dependent Care Part in the Travel Policy covers $40 per day for care for my child while I'm away—can you let me know how I claim this for the rollaway bed (my partner and son are travelling to stay with family while I'm away, so it's a cost over and above my usual care arrangement). I can't see anywhere on the travel form to include it.* (25th July 2019, email from participant P2 to (University) Travel)

*"The Dependant (sic) Care Allowance is only available for staff where they have to make an alternative arrangement to their 'normal' care for a dependant (sic) i.e. extra day care, a relative to look after them etc. In your case this would not apply as you and your husband will still be with your child and the additional cost for a rollaway is a personal cost to you.* (Response from (University) Travel)

In contrast, another participant reported that their claim for higher hotel accommodation costs because their child and partner were travelling with them was approved automatically.

This last interaction illustrates that the policy is administered within an existing framework of gendered language and assumptions, where a "partner" of a female employee was assumed to be a male husband. It was also assumed that employee would always have a partner available to care for a dependent. Organisational language reinforces unconscious bias and contributes to cultural norms. Gendered language at a national level has been linked to lower levels of equality (Prewitt-Freilino et al. 2012), while Evans (2019) identified the impact of "benevolent sexism" on women's progress and the damage done by perceptions by male managers regarding aspirations of women for managerial roles and family priorities.

*3.2. Organisational Culture and an Unfunded Policy*

Travel is a condition of employment at the study university, given its distributed campus model. Most staff employment contracts include a clause that can require travel of up to 500 kms per day. The university commits to funding this travel through a policy statement, and through travel budgets within each functional area.

*It is the University's policy to reimburse employees for ordinary, necessary and reasonable travel expenses that are directly connected with or pertaining to the conduct of* [University] *business.*

*Employees are expected to exercise prudent business judgment regarding expenses covered by this policy. They are expected neither to gain nor lose financially when travelling on* [University] *business."*

The Dependent Care Support policy is an unfunded policy at a university level, which means that expense claims are funded by individual research or professional budgets, or a divisional account. For researchers, this creates issues in developing budgets for external grant applications, as many grant conditions explicitly exclude caring costs from the application process. Participants' comments below illustrate the challenges that arise because there is no clear pathway for funding this policy.

For example, when discussing a recent Dependent Care Support expense claim, one participant reflected on the fact the payment came from a research account, but that the expense hadn't been included in the original grant budget:

*We are both on the same project, it is externally funded ( … ) and subcontracted through another University. ( … ) We both know that in the initial budget there was no such cost* [dependent care] *budgeted for.*

*So how does this make me feel? Unsatisfied. Still feeling a bit like fraud. I know I am doing the right thing, I know I was 'lucky' to have this approved, however, I also know that there is no way it should have been approved the way it was because these funds are now coming from something else that they were allocated to.* (P6)

For managers, tight budgets in a corporate university environment raise issues of equity and opportunity to travel:

*We are already tight on funding in so many areas. $40 might not sound like much but if you have an entire team eligible and these extra payments with their travel wasn't accounted in to the budget for at the start of the year it could impact the bottom line. On one hand you want to tell everyone to apply, on the other hand you really want to hide it because where would the money come from if they did all ask for it multiple times a year?* (P4)

P6 *stated this is an unfunded policy. This makes me so cranky. What is the point of the policy? Service areas simply don't have the money to fund it. It feels like lip-service from the uni with no backing. What do we do when we literally don't have the money to pay but the uni states there is a policy so we have to pay? How is this equitable?* (P4)

Staff making claims for expenses also had affective reactions to their claims:

[I] *was successful in getting dependent care. Why do I feel the need to not actually follow through and claim the money when I return? I know how my supervisor feels about this policy and that the money comes from our research budget and so I should not claim it. What would make this feel normal and ok about this?* (P4)

*I was so rushed filling out the form I didn't have the time to worry about the money and where it was coming from, but now I'm starting to worry it will be questioned, as the* [budget manager] *actually questioned someone else's travel plan in that meeting [ … ] it all felt a bit intensely interrogated and public.* (P2)

By requiring individual budget holders to 'find room' in their budget to support Dependent Care Support expense claims, any travel done by staff with dependent care responsibilities for work reasons will by necessity have a higher budget requirement. This potentially creates opportunity costs to those who would be most likely to benefit from the expense reimbursements—employees (mostly women) with dependent care responsibilities.

### 3.3. Policy Development and Implementation

Based on the data presented in Table 1, there seems to be minimal awareness across the university about the policy and its associated processes. Given that travel forms part of employment conditions for university staff, it is surprising that the policy is not better publicised, with a clearer application process accessible to all.

*How are employees supposed to find out about this policy? When we asked a room full of 25* [female employees of the university], *none of them had heard of the policy. The majority of* [the women] *have children, and all of whom have had to travel for the* [meeting]. *In my first few months at* [the university] *I had to travel to a conference. I was still breastfeeding my youngest child and had to make arrangements to bring him with me as well as my mother to care for him. Both my supervisor and the* [travel booking office] *obviously knew I was arranging care for my dependent child but no one mentioned this policy to me. This trip cost me around $500 in travel and car hire costs.* (P1)

*I had two senior staff members tell me they only just learnt about the policy either this year or last and they wish they had known when their kids where young as it would have really helped.* (P4)

*It's still not clear what makes you eligible to claim the allowance and for what, so this will be very interesting to see how it goes. I still have never been asked to prove my child's age, which is actually the only thing the policy says they can do!* (P2)

*I also tried to touch on the subject with my current supervisor, who is part-time on the same project as well as someone who has dependents and does frequent travel. Nonresponse so far to my e-mail on this, which makes me feel like it's a topic that's been avoided, or too complex to debate or tried to analyze. I am sure there are many more staff in this same position of uncertainty, whether they are staff or managers.* (P6)

In the 10 years since its introduction, the Dependent Care Support policy terms have been updated twice. This has resulted in the original intended outcomes of the policy being further restricted, and raises questions of the role of policy in determining the definition of dependent care and its cost.

When first introduced, the policy simply referred to 'dependent care', without defining who dependents were, allowing claims to be made in situations of child, disabled and elderly care. In 2012, the policy was revised to say that the support applies to 'dependents up to the age of 16 years, or where the dependent is in receipt of a disability support'. This had the potential to have a substantial impact on a diverse range of care-givers and appeared to occur without stakeholder engagement or consultation.

In 2019, the policy was amended again so that costs associated with caring responsibilities for children under the age of 16 were the only expense able to be claimed. It is no longer policy to support claims for the dependent care of people over the age of 16 in receipt of a disability payment. This was accompanied by a statement that reported that no significant changes had been made to the policy.

Defining 'dependent' and 'dependent care expenses' was key to most participants, as the policy did not provide enough detail for managing staff to make decisions:

> *Is it appropriate to claim the entitlement for a partner to care for the child? The partner may be taking leave without pay to care for dependents.* (P5)

> *One of my children has autism which makes the demands of parenting even harder. It is not a simple situation of just finding someone or somewhere to provide care when I am away. The energy it would require for her to be cared for out of her usual routine would be considerable and would impact her for several days. From talking to other parents and managers throughout this project, I don't think managers often consider the many complex aspects of caring for dependents that can be involved and how it is not just as simple as getting a babysitter.* (P1)

Towards the end of this project it was discovered that the allowance is liable for Fringe Benefits Tax after $2000 (~50 days of travel), which will have a significant impact on some staff members' taxable income and family allowances including Child Care Subsidy entitlement, especially given child care is not an allowable tax deduction. It is important that these wider contexts be taken into account when considering the impact of work-related travel on families.

### 3.4. Gender Equity

This study has shown that the successful uptake of this policy is low and to date, has been used solely by women. Given that this policy has been highlighted in the university's application for an Athena SWAN Award, is acknowledged in the award of Employer of Choice for Gender Equality from the Commonwealth Workplace Gender Equality Agency (WGEA), and is also highlighted in the university's Workplace Gender Equity Strategy, understanding how this policy is being successfully taken up and by whom is important. For example,

> *It just won't help. What's the point in applying for that amount? I just can't afford to travel do the [work]. I feel guilty that others in my team have to do it instead but I can't cover the cost of childcare.* (P8)

> *I feel there is no recognition given to the fact that travel, or any work required outside of standard hours, has a large impact on the lives of employees. People use this time to spend with family and care for others but also be part of communities, for exercise and self-care, volunteer, look after their animals and properties. By not recognising that when travelling or working evenings, employees commit 16+ hours to work, the university is not recognising the important roles of their employees outside of* [the university]. *I feel that this attitude comes from a culture of when men worked and women stayed home, to care for children but also care for the elderly, sick and disabled, volunteer in the community, do the housework and bear the mental load of running a household. For gender equity to be achieved we need to recognise the value of the roles of everyone outside of their paid employment. By requiring men and women to be available to travel without addressing the additional hours as well as financial burden of this, gender equity for women is not going to be achieved as women still bear the majority of*

*these roles. By not supporting men in this way it makes it difficult for their partners to work. There is an assumption that there is a primary care giver rather than two parents raising children together.* (P1)

The potential for the policy to actually reinforce gender stereotypes relating to care must also be considered, either by strengthening existing gendered organisational culture (Acker 2012) failing to address the underlying issues about definitions of jobs (Grummell et al. 2009). It is therefore equally important to track unsuccessful applications to examine how the policy is being used by all genders.

We know that gender stereotypes are pervasive in our society and that the role of caregiving more often than not falls to women. This includes caregiving responsibilities—not just for children, but for adult dependents with a disability and aging parents. This directly impacts women's ability and choice to travel for work, and consequently women are being required to make a choice between their caring responsibilities and their career development. This study found examples of people unable to travel because of the cost involved and this may have ramifications for promotion and future employment opportunities. These stereotypes certainly were the reality for most of the participants with caring responsibilities in this study. In some ways, it appeared easier for participants not to interact with the policy because of the extra energy required for the outcome.

## 4. Discussion

This research project developed four recommendations based on themes emerging from the study data analysis, aimed at enhancing the capability of the policy to further enhance the efforts being made to work towards gender equity in the study institution. The findings, while specific to a single university, provide key discussion points for consideration by all institutions working to this aim, and scaffolding for reflection on their practices and goals.

*Creating Consistent, Transparent Application Processes*

This study has highlighted the need for a clear, transparent process for the administration of gender equity policies, including claims for Dependent Care Support expenses. University administration and support areas need to be fully supported in their capacity to administer these policies in a way that is clear to them, and also to the applicants. This includes but is not limited to, travel offices, budget managers, research office staff and senior leadership. This transparency also needs to include a mechanism for tracking and monitoring unsuccessful applications, as well as a process for dispute resolution.

*Improving Awareness and Communication*

The study identified that one major weakness of this university's Dependent Care Support policy was the lack of communication about its existence and how to manage and apply for it. For all institutions it is recommended that clear internal communication strategies are developed that target both those eligible to claim expenses, along with those administering the policy. It is also recommended that induction packages for new staff and managers highlight any specific gender equity policies and put them in the context of the organisation's gender equity agenda. Underpinning these strategies should be approaches that make childcare responsibilities for all staff more visible, regardless of gender (Grummell et al. 2009, p. 197)

*Policy Review Processes*

Underpinning the application process and communication is the need to address the underlying organisational issues that include the impact of how university work is assessed and defined, with employer expectations aligning with a "care-less" employee (Grummell et al. 2009, p. 192). This organisationally ideal but nonexistent employee encourages all genders to render their caring responsibilities invisible (Grummell et al. 2009, p. 192). This "strategic disappearance" (Lewis and Simpson 2010, p. 13) may explain the non-uptake of this policy by males with caring responsibilities and certainly contributes to poor uptake overall. This is particularly significant because of the absence of any mechanism to track policy applications, regardless of the outcome. The fact that the amount that can be claimed under the travel policy has not increased in value over the last 30 years also points to institutional reluctance for self-reflection (Butterwick and Dawson 2005, p. 53). It is recommended that

gender equity policies be evaluated regularly using both quantitative and qualitative data, including developing and improving data collection and analysis techniques. This would allow for, for example, the consideration of the real needs of those with dependents, including the cost of care and the definition of dependent. This could be achieved through annual reviews of professional and academic staff employment data, and through qualitative research, the specifics of staff caring responsibilities, including children, aging relatives and those with disabilities. These reviews should also include quantification of unsuccessful applications for any policies relating to gender equity. This information can also inform reviews of the policy application and benchmarking activities.

*High Level Policy Funding*

Policies such as a Dependent Care Support policy are major contributors towards achieving gender equity within an institution. The particular policy studied in this paper is one of the only policies of its kind within Australian universities and sets a standard for other universities and similar distributed institutions. The future success of policies such as this in achieving gender equity lies in, as a core feature, organisational level funding of policy implementation. This would speak to the value that the university places on supporting caring responsibilities while also maintaining employment opportunities. Without funding, institutions risk perpetuating modes of discrimination based on dependent care responsibilities, as budget holders make decisions to fund travel with their individual budget bottom line in mind. By funding gender equity policies at a university level, this pressure is no longer apparent, and care becomes a visible part of staff performance (Grummell et al. 2009, p. 204; Lewis and Simpson 2010).

The study also provided an opportunity to examine the experiences of a cross-section of employees from a regional Australian university. While the sample is small, the similarities of experiences of academic staff with caring responsibilities, and who are reliant on research contract employment are significant. Other similarities include the difficulty in implementing the policy by staff with management roles, whether at a professional or academic level. This speaks to the organisational gendered subculture that ensures that the burden of caring responsibilities remains invisible and unfunded.

## 5. Conclusions

This study has highlighted an innovative policy that is working to redress gender inequity. Effective ongoing review of policy implementation and funding processes will support those with caring responsibilities to effectively engage in their work life without the additional financial burden that is currently experienced. This is particularly pertinent for a distributed higher education institution with considerable travel requirements; however, all higher educational providers who wish to reduce the gender equity gap would be encouraged to introduce similar policies and learn from this study in ensuring effective implementation.

Further research is required in this area, including a more in-depth investigation into the spectrum of staff the policy affects (those who have not applied and those who have applied unsuccessfully). Additional research is also recommended to explore the impact of current global austerity measures in higher education and the impact these have on gender equity.

**Author Contributions:** Conceptualization, J.M., J.L., G.R., K.M.-O., S.C. and C.T.; methodology, J.L., J.M. and G.R.; formal analysis, J.M., J.L., G.R., K.M.-O. and S.C.; data curation, J.M.; writing—original draft preparation, J.M.; writing—review and editing, J.L., G.R., K.M.-O., S.C. and C.T.; project administration, J.M. All authors have read and agree to the published version of the manuscript.

**Funding:** The authors acknowledge and thank the Tri-Faulty Open Access Scheme, Charles Sturt University, for assistance with publishing costs.

**Acknowledgments:** The authors thank their mentor on this research project, Cate Thomas, for her support and encouragement. The authors also thank Hedy Bryant and Marcelle Droulers for their support during the authors' participation in a Leadership Development for Women program in 2019.

**Conflicts of Interest:** The authors declare no conflict of interest.

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
