# Peer review of "The Lived Experience of Gender and Gender Equity Policies at a Regional Australian University"

_socsci, doi:10.3390/socsci9070115_

Round 1

Reviewer 1 Report

The issue of the article – the discrepancies between a gender equality policy and the lived experiences – is vital to understand better the barriers to implement gender equality in higher education. The case study (an Australian university) analyses a policy by which employees who are traveling for work may get refunding for additional care-related expenses. The methods, statistical analysis of administrative data, and collaborative auto-ethnography are appropriate to the research question. The results are essential, showing gatekeeping by the administration, organisational and affective barriers because funds do not cover the policy, limitations of the policy in the course of times, and gendered effects of the policy.

Nevertheless, I have some remarks to improve the article.

Mainly, the study is suffering from the lack of a theoretical framework. The authors might use theoretical approaches like “Gendered organisation” (Acker), “Care regimes” (Pocock) or analysing gender and diversity programs as “window dressing” (Williams et al. 2015; Ahmed) to frame their study. This might help to sharpen the argumentation, to use more lucid notions and to link the result better to the scientific discussion. Especially the discussion is lacking a theoretical framework. The discussion consists only of recommendations for practitioners. These recommendations are useful, but a scientific article should also discuss how the results contribute to the scientific discussion.

Furthermore, the part “gender equity” could be improved by a theoretical framework. The chapter ends with a long citation without analysis. Instead, the gendered nature of the policy – only women received payments – and possible contradictory effects like strengthening gendered work and care arrangements and gender stereotypes should be analysed more in detail. Finally, I propose an analytical heading instead of the ironic statement “continuous improvement”.

Concerning the methods, the total number of participants in the study is lacking. The variables should be more structured (one variable on dependent care responsibilities with the categories yes and no instead of two variables on this issue, same for long-serving staff and recent appointments).

Because of the lack of a theoretical framework, I ask for major revisions.

Reviewer 2 Report

The paper describes the implementation of a specific gender equity policy in the context of an Australian university. The policy allows people with care responsibilities to apply for costs to arrange child care while travelling. Based on a mixed method approach the policy describes the take up of the policy and problems regarding its implementation. The analysis remains rather descriptive and therefore unsatisfactory for readers.

The title of the paper announces the discussion of experiences with gender and gender equity policies. However, it is not clear how gender equity is defined and why the focus is on gender equity and gender equality. The authors neither describe the gender equity (equality?) goal of the policy or the institution. They mention that the policy is part of the Athena Swan application of the institution. Hence, the policy of the institution should aim at gender equality not just gender equity. If the concrete policy aims at gender equity (as part of the overall gender equality approach) this should be explicated. However, it is necessary to embed the concrete policy in the overall policy mix.

Furthermore, it would be necessary to strengthen the gender dimension of the policy and its implementation in the analysis and discussion. The analysis makes clear that it is in fact a woman only policy which addresses them in their role of potential care givers. The analysis should be complemented by a more critical discussion of the associated gender aspects of the policy. This could be done by including a critical reflection of the perception of gatekeepers, the non-take up by people eligible for the benefits or an analysis of documents related to the policy (e.g. how the goal of the policy or the target group is defined).

The quantitative data presented covers 14 cases of successful applications. This figures alone do not tell much about the take up of the policy. The analysis could be enriched by including information about refused applications (including the reasons for rejection). However, even more important would be an analysis of non-take up. This would require to focus on people eligible who do not apply for the support. The qualitative information depicts a lack of information about the policy which is probably one reason for non-take up. The non-take up might also reflect other gendered problems in academia (e.g. the assignment of care responsibilities to the private life).

Regarding the qualitative data presented it is not clear how many cases are included in the analysis (five research team members plus others). Additional information about the selection process of cases would be helpful for readers.

Reviewer 3 Report

This work on experiences of gender and gender equity within higher education is certainly a necessary point of engagement.  On a general level, the manuscript provides an interesting take on the experiences of women academics and their experiences with an institutional childcare policy, specifically as it relates to access, relevance and impact on the scholarly activities.  However, on a more specific level, there are some gaps that can be strengthened to make the work even more impactful. These include:

  1. Argument on Gendered experiences-Some interrogation first of the gendered experiences of these women academics within higher education. If there is no primary data to speak to these issues, then perhaps a recommendation would be for the use of empirical literature to provide the necessary foundation that echoes or reinforces the argument on the gendered experiences. At present, this rationale or justification or argumentative position is missing from the paper. 
  2.  Clarification of the methodology-While mixed methods are presented as a sources of data for the study, there is the interchangeable use of research design with the method, in this case, between collaborative autoethnography as a research design and quantitative secondary data as the method. A closer examination shows the use of multiple methods around journaling, observations, conversations, etc. Thus, some clarity around this would be useful.  
  3. Selection of Participants-In the identification of the participants, there is reference to participants who have no dependents and participants with dependents. Similarly, there is an indication that participants included recent employees and other with over ten years. While these are stated, there is a lack of information on the rationale for these categories of participants. Additionally, there is little, if any, comparative analyses of the findings related to these contrasting groups. Some clarity on this issue would also enhance the paper. 
  4. Presentation of findings-similarly, there is evidence in the paper that participants included authors of this manuscript and other women academics within the institution. However, there is no distinction in the presentation and analysis of the findings as it relates to these two groups. Questions that emerge therefore are: were there any distinguishable differences between how these women experienced or engaged with the policy? What were the level of sensitization for women between these groups, and more importantly, how was the policy perceived by these different groups? 
  5.  Data analysis-In the analysis of the data, there is a suggestion that the gendered language of the policy may be a significant factor in how one makes sense of the link between the experiences of these women and the policy itself. This gendered language around the policy however, while alluded to, is not elaborated or presented within the paper. It is necessary therefore to substantiate that argument. 
  6. Discussion- While the focus of the paper was on the experiences of women academics, given the data on access, there is a need to tease out the factors that inadvertently lead to this 'gendered reality' around access to the facility. Thus, an important question to consider is that of why is it that male academics have not accessed this facility? Can one assume in this case this is due to the fact established earlier in the paper that women bear the burden for childcare? What then of the potential situation where there is a male academic who is also a single father? While that may be in the minority, (assuming here), the lack of access to the facility by such a group, is a matter for a gendered analysis. This has been omitted from the paper and can be addressed.
  7. Methodological thrust-While the collaborative auto-ethnographic design is presented here, there is a lack of analysis of the socio-cultural facets of these experiences that are consistent with this methodological thrust. Thus, another question is that of what were the mechanisms of support and familial and social realities of these women? At this point, the data delves into to an institutional reality, which lack needed depth on the relational experiences that affect their experiences with child care and academic work. Attending to these can significantly enhance the paper. 

Round 2

Reviewer 1 Report

The paper has improved by explaining the methods in more detail. Framing the study by some theoretical references (Acker, Grummell), the argumentation is more grounded, as far as this is possible without rewriting the paper. I understand that the paper is part of academic institutional reflection. This framing helps to understand the conclusion, written in the form of recommendations.

The study results are very interesting: administrative gatekeepers, gendered language, a gender-neutral policy taken only by women, contraction between highlighting the policy in the Athena SWAN application, and the non-funding. The results ware worth being published now. But I recommend working further on the issue, with a more profound theoretical framework and more data (non-applications like suggested by other reviewers).

The paper needs some text editing (errors after rewriting).

Reviewer 2 Report

Thank you for revising that paper. I appreciate that most comments have been addressed in the revision. The paper is more convincing compared to the frist version but it is still rather descriptive and its significance remains limited due to the weak link to institutional equity policies. What has not been done is to embed the policy in the overall gender equity policy mix. What the paper provides an indepth look in the experiences of women applying for the funding. This is a rather limited focus of the paper. Due to the data restrictions it is not possible to analyse the share of successful implementations. This is a pity but limits the scope of the paper too.

A minor comment: page two it should be "drawing upon" instead of "drawing onon"

Author Response

Thanks for your comments. We do hope to be able to address the data deficit in the future. With the raised awareness of this policy stemming from this project, we hope to be able to expand the research for a more in-depth look at gender equity at this institution.

Thanks also for the correction.

Reviewer 3 Report

This is a significant improvement from the first draft. The paper also flows well with relevant additions/amendments made. Authors have substantively addressed the concerns raised in the first review. There are a few aspects of the paper that are in need of refinement. These include:

  1. In speaking to the methods, the authors mention the use of journals, observations, institutional data, and conversations. However, outside of the institutional data, which can be clearly identified, there is no distinction in the sources of the data used in the reporting of the qualitative data. Perhaps a way out here would be to revised the methods to include the use of institutional data including emails and a conversation between collaborators. 
  2. A minor suggestion here is to change the use of 'multiple variables' used here to define participants and perhaps to use 'diverse profiles' or an equivalent. This appears just before the use of the table to describe participants. The use of variable here can be mistakenly interpreted within the context of quantitative research. 
  3. finally, a recommendation here to edit to remove double wording, repeated full stops or commas, inserting page numbers where direct quotations are used, for instance in the definition of gender equality etc.
  4. Overall, great revision of the paper. 
